# Online control of the false discovery rate with decaying memory

**Aaditya Ramdas**    **Fanny Yang**    **Martin J. Wainwright**    **Michael I. Jordan**
University of California, Berkeley
{aramdas, fanny-yang, wainwrig, jordan} @berkeley.edu

## Abstract

In the online multiple testing problem, $p$-values corresponding to different null hypotheses are observed one by one, and the decision of whether or not to reject the current hypothesis must be made immediately, after which the next $p$-value is observed. Alpha-investing algorithms to control the false discovery rate (FDR), formulated by Foster and Stine, have been generalized and applied to many settings, including quality-preserving databases in science and multiple A/B or multi-armed bandit tests for internet commerce. This paper improves the class of generalized alpha-investing algorithms (GAI) in four ways: (a) we show how to uniformly improve the power of the entire class of monotone GAI procedures by awarding more alpha-wealth for each rejection, giving a win-win resolution to a recent dilemma raised by Javanmard and Montanari, (b) we demonstrate how to incorporate prior weights to indicate domain knowledge of which hypotheses are likely to be non-null, (c) we allow for differing penalties for false discoveries to indicate that some hypotheses may be more important than others, (d) we define a new quantity called the decaying memory false discovery rate (mem-FDR) that may be more meaningful for truly temporal applications, and which alleviates problems that we describe and refer to as "piggybacking" and "alpha-death." Our GAI++ algorithms incorporate all four generalizations simultaneously, and reduce to more powerful variants of earlier algorithms when the weights and decay are all set to unity. Finally, we also describe a simple method to derive new online FDR rules based on an estimated false discovery proportion.

## 1  Introduction

The problem of multiple comparisons was first recognized in the seminal monograph by Tukey [12]: simply stated, given a collection of multiple hypotheses to be tested, the goal is to distinguish between the nulls and non-nulls, with suitable control on different types of error. We are given access to one $p$-value for each hypothesis, which we use to decide which subset of hypotheses to reject, effectively proclaiming the rejected hypothesis as being non-null. The rejected hypotheses are called *discoveries*, and the subset of these that were truly null—and hence mistakenly rejected—are called *false discoveries*. In this work, we measure a method's performance using the *false discovery rate* (FDR) [2], defined as the expected ratio of false discoveries to total discoveries. Specifically, we require that any procedure must guarantee that the FDR is bounded by a pre-specified constant $\alpha$.

The traditional form of multiple testing is *offline* in nature, meaning that an algorithm testing $N$ hypotheses receives the entire batch of $p$-values $\{P_1, \ldots, P_N\}$ at one time instant. In the *online* version of the problem, we do not know how many hypotheses we are testing in advance; instead, a possibly infinite sequence of $p$-values appear one by one, and a decision about rejecting the null must be made before the next $p$-value is received. There are at least two different motivating justifications for considering the online setting:

M1. We may have the entire batch of $p$-values available at our disposal from the outset, but we may nevertheless choose to process the $p$-values one by one in a particular order. Indeed, if one can use prior knowledge to ensure that non-nulls typically appear earlier in the ordering, then carefully designed online procedures could result in more discoveries than offline algorithms (that operate without prior knowledge) such as the classical Benjamini-Hochberg algorithm [2], while having the same guarantee on FDR control. This motivation underlies one of the original online multiple testing paper, namely that of Foster and Stine [5].

M2. We may genuinely conduct a sequence of tests one by one, where both the choice of the next null hypothesis and the level at which it is tested may depend on the results of the previous tests. Motivating applications include the desire to provide anytime guarantees for (i) internet companies running a sequence of A/B tests over time [9], (ii) pharmaceutical companies conducting a sequence of clinical trials using multi-armed bandits [13], or (iii) quality-preserving databases in which different research teams test different hypotheses on the same data over time [1].

The algorithms developed in this paper apply to both settings, with emphasis on motivation M2.

Let us first reiterate the need for corrections when testing a sequence of hypotheses in the online setting, even when all the p-values are independent. If each hypothesis $i$ is tested independently of the total number of tests either performed before it or to be performed after it, then we have no control over the number of false discoveries made over time. Indeed, if our test for every $P_i$ takes the form $\mathbf{1}\{P_i \leq \alpha\}$ for some fixed $\alpha$, then, while the type 1 error for any individual test is bounded by $\alpha$, the set of discoveries could have arbitrarily poor FDR control. For example, under the "global null" where every hypothesis is truly null, as long as the number of tests $N$ is large and the null $p$-values are uniform, this method will make at least one rejection with high probability (w.h.p.), and since in this setting every discovery is a false discovery, w.h.p. the FDR will equal one.

A natural alternative that takes multiplicity into account is the Bonferroni correction. If one knew the total number $N$ of tests to be performed, the decision rule $\mathbf{1}\{P_i \leq \alpha/N\}$ for each $i \in \{1, \ldots, N\}$ controls the probability of even a single false discovery—a quantity known as the familywise error rate or FWER—at level $\alpha$, as can be seen by applying the union bound. The natural extension of this solution to having an unknown and potentially infinite number of tests is called *alpha-spending*. Specifically, we choose any sequence of constants $\{\alpha_i\}_{i \in \mathbb{N}}$ such that $\sum_i \alpha_i \leq \alpha$, and on receiving $P_i$, our decision is simply $\mathbf{1}\{P_i \leq \alpha_i\}$. However, such methods typically make very few discoveries—meaning that they have very low power—when the number of tests is large, because they must divide their error budget of $\alpha$, also called alpha-wealth, among a large number of tests.

Since the FDR is less stringent than FWER, procedures that guarantee FDR control are generally more powerful, and often far more powerful, than those controlling FWER. This fact has led to the wide adoption of FDR as a de-facto standard for offline multiple testing (note, e.g., that the Benjamini-Hochberg paper [2] currently has over 40,000 citations).

Foster and Stine [5] designed the first online alpha-investing procedures that use and earn alpha-wealth in order to control a modified definition of FDR. Aharoni and Rosset [1] further extended this to a class of generalized alpha-investing (GAI) methods, but once more for the modifed FDR. It was only recently that Javanmard and Montanari [9] demonstrated that monotone GAI algorithms, appropriately parameterized, can control the (unmodified) FDR for independent $p$-values. It is this last work that our paper directly improves upon and generalizes; however, as we summarize below, many of our modifications and generalizations are immediately applicable to all previous algorithms.

**Contributions and outline.** Instead of presenting the most general and improved algorithms immediately, we choose to present results in a bottom-up fashion, introducing one new concept at a time so as to lighten the symbolic load on the reader. For this purpose, we set up the problem formally in Section 2. Our contributions are organized as follows:

1. **Power.** In Section 3, we introduce the generalized alpha-investing (GAI) procedures, and demonstrate how to uniformly improve the power of monotone GAI procedures that control FDR for independent $p$-values, resulting in a win-win resolution to a dilemma posed by Javanmard and Montanari [9]. This improvement is achieved by a somewhat subtle modification that allows the algorithm to reward more alpha-wealth at every rejection but the first. We refer to our algorithms as *improved generalized alpha-investing* (GAI++) procedures, and provide intuition for why they work through a general super-uniformity lemma (see Lemma 1 in Section 3.2). We

also provide an alternate way of deriving online FDR procedures by defining and bounding a natural estimator for the false discovery proportion $\widehat{\text{FDP}}$.

2. **Weights.** In Section 5, we demonstrate how to incorporate certain types of prior information about the different hypotheses. For example, we may have a prior weight for each hypothesis, indicating whether it is more or less likely to be null. Additionally, we may have a different penalty weight for each hypothesis, indicating differing importance of hypotheses. These prior and penalty weights have been incorporated successfully into offline procedures [3, 6, 11]. In the online setting, however, there are some technical challenges that prevent immediate application of these offline procedures. For example, in the offline setting all the weights are constants, but in the online setting, we allow them to be random variables that depend on the sequence of past rejections. Further, in the offline setting all provided weights are renormalized to have an empirical mean of one, but in the truly online setting (motivation M2) we do not know the sequence of hypotheses or their random weights in advance, and hence we cannot perform any such renormalization. We clearly outline and handle such issues and design novel prior- and/or penalty-weighted GAI++ algorithms that control the penalty-weighted FDR at any time. This may be seen as an online analog of doubly-weighted procedures for the offline setting [4, 11]. Setting the weights to unity recovers the original class of GAI++ procedures.

3. **Decaying memory.** In Section 6, we discuss some implications of the fact that existing algorithms have an infinite memory and treat all past rejections equally, no matter when they occurred. This causes phenomena that we term as "piggybacking" (a string of bad decisions, riding on past earned alpha-wealth) and "alpha-death" (a permanent end to decision-making when the alpha-wealth is essentially zero). These phenomena may be desirable or acceptable under motivation M1 when dealing with batch problems, but are generally undesirable under motivation M2. To address these issues, we propose a new error metric called the *decaying memory false discovery rate*, abbreviated as mem-FDR, that we view as better suited to multiple testing for truly temporal problems. Briefly, mem-FDR pays more attention to recent discoveries by introducing a user-defined discount factor, $0 < \delta \leq 1$, into the definition of FDR. We demonstrate how to design GAI++ procedures that control online mem-FDR, and show that they have a stable and robust behavior over time. Using $\delta < 1$ allows these procedures to slowly forget their past decisions (reducing piggybacking), or they can temporarily "abstain" from decision-making (allowing rebirth after alpha-death). Instantiating $\delta = 1$ recovers the class of GAI++ procedures.

We note that the generalizations to incorporate weights and decaying memory are entirely orthogonal to the improvements that we introduce to yield GAI++ procedures, and hence these ideas immediately extend to other GAI procedures for non-independent $p$-values. We also describe simulations involving several of the aforementioned generalizations in Appendix C.

## 2 Problem Setup

At time $t = 0$, before the $p$-values begin to appear, we fix the level $\alpha$ at which we wish to control the FDR over time. At each time step $t = 1, 2, \ldots$, we observe a $p$-value $P_t$ corresponding to some null hypothesis $H_t$, and we must immediately decide whether to reject $H_t$ or not. If the null hypothesis is true, $p$-values are stochastically larger than the uniform distribution ("super-uniform", for short), formulated as follows: if $\mathcal{H}^0$ is the set of true null hypotheses, then for any null $H_t \in \mathcal{H}^0$, we have

$$\Pr\{P_t \leq x\} \leq x \quad \text{for any } x \in [0, 1]. \tag{1}$$

We do not make assumptions on the marginal distribution of the $p$-values for hypotheses that are non-null / false. Although they can be arbitrary, it is useful to think of them as being stochastically smaller than the uniform distribution, since only then do they carry signal that differentiates them from nulls. Our task is to design threshold levels $\alpha_t$ according to which we define the rejection decision as $R_t = \mathbf{1}\{P_t \leq \alpha_t\}$, where $\mathbf{1}\{\cdot\}$ is the indicator function. Since the aim is to control the FDR at the fixed level $\alpha$ at any time $t$, each $\alpha_t$ must be set according to the past decisions of the algorithm, meaning that $\alpha_t = \alpha_t(R_1, \ldots, R_{t-1})$. Note that, in accordance with past work, we require that $\alpha_t$ does not directly depend on the observed $p$-values but only on past rejections. Formally, we define the sigma-field at time $t$ as $\mathcal{F}^t = \sigma(R_1, \ldots, R_t)$, and insist that

$$\alpha_t \in \mathcal{F}^{t-1} \quad \equiv \quad \alpha_t \text{ is } \mathcal{F}^{t-1}\text{-measurable} \quad \equiv \quad \alpha_t \text{ is } \textit{predictable}. \tag{2}$$

As studied by Javanmard and Montanari [8], and as is predominantly the case in offline multiple testing, we consider *monotone* decision rules, where $\alpha_t$ is a coordinate-wise nondecreasing function:

$$\text{if } \tilde{R}_i \geq R_i \text{ for all } i \leq t - 1, \text{ then we have } \alpha_t(\tilde{R}_1, \ldots, \tilde{R}_{t-1}) \geq \alpha_t(R_1, \ldots, R_{t-1}). \tag{3}$$

Existing online multiple testing algorithms control some variant of the FDR over time, as we now define. At any time $T$, let $R(T) = \sum_{t=1}^{T} R_t$ be the total number of rejections/discoveries made by the algorithm so far, and let $V(T) = \sum_{t \in \mathcal{H}^0} R_t$ be the number of false rejections/discoveries. Then, the false discovery proportion and rate are defined as

$$\text{FDP}(T) := \frac{V(T)}{R(T)} \quad \text{and} \quad \text{FDR}(T) = \mathbb{E}\left[\frac{V(T)}{R(T)}\right],$$

where we use the dotted-fraction notation corresponds to the shorthand $\frac{a}{b} = \frac{a}{b \vee 1}$. Two variants of the FDR studied in earlier online FDR works [5, 8] are the *marginal* FDR given by $\text{mFDR}_\eta(T) = \frac{\mathbb{E}[V(T)]}{\mathbb{E}[R(T)]+\eta}$, with a special case being $\text{mFDR}(T) = \frac{\mathbb{E}[V(T)]}{\mathbb{E}[R(T) \vee 1]}$, and the *smoothed* FDR, given by $\text{sFDR}_\eta(T) = \mathbb{E}\left[\frac{V(T)}{R(T)+\eta}\right]$. In Appendix A, we summarize a variety of algorithms and dependence assumptions considered in previous work.

## 3 Generalized alpha-investing (GAI) rules

The generalized class of alpha-investing rules [1] essentially covers most rules that have been proposed thus far, and includes a wide range of algorithms with different behaviors. In this section, we present a uniform improvement to monotone GAI algorithms for FDR control under independence.

Any algorithm of the GAI type begins with an *alpha-wealth* of $W(0) = W_0 > 0$, and keeps track of the wealth $W(t)$ available after $t$ steps. At any time $t$, a part of this alpha-wealth is used to test the $t$-th hypothesis at level $\alpha_t$, and the wealth is immediately decreased by an amount $\phi_t$. If the $t$-th hypothesis is rejected, that is if $R_t := \mathbf{1}\{P_t \leq \alpha_t\} = 1$, then we award extra wealth equaling an amount $\psi_t$. Recalling the definition $\mathcal{F}^t := \sigma(R_1, \ldots, R_t)$, we require that $\alpha_t, \phi_t, \psi_t \in \mathcal{F}^{t-1}$, meaning they are predictable, and $W(t) \in \mathcal{F}^t$, with the explicit update $W(t) := W(t-1) - \phi_t + R_t\psi_t$. The parameters $W_0$ and the sequences $\alpha_t, \phi_t, \psi_t$ are all user-defined. They must be chosen so that the total wealth $W(t)$ is always non-negative, and hence that $\phi_t \leq W(t-1)$ If the wealth ever equals zero, the procedure is not allowed to reject any more hypotheses since it has to choose $\alpha_t$ equal to zero from then on. The only real restriction for $\alpha_t, \phi_t, \psi_t$ arises from the goal to control FDR. This condition takes a natural form—whenever a rejection takes place, we cannot be allowed to award an arbitrary amount of wealth. Formally, for some user-defined constant $B_0$, we must have

$$\psi_t \leq \min\{\phi_t + B_0, \frac{\phi_t}{\alpha_t} + B_0 - 1\}. \tag{4}$$

Many GAI rules are not monotone (cf. equation (3)), meaning that $\alpha_t$ is not always a coordinatewise nondecreasing function of $R_1, \ldots, R_{t-1}$, as mentioned in the last column of Table 2 (Appendix A). Table 1 has some examples, where $\tau_k := \min_{s \in \mathbb{N}} \mathbf{1}\left\{\sum_{t=1}^{s} R_t = k\right\}$ is the time of the $k$-th rejection.

| Name | Parameters | Level $\alpha_t$ | Penalty $\phi_t$ | Reward $\psi_t$ |
|------|-----------|------------------|------------------|-----------------|
| [5] Alpha-investing (AI) | — | $\frac{\phi_t}{1+\phi_t}$ | $\leq W(t-1)$ | $\phi_t + B_0$ |
| [1] Alpha-spending with rewards | $\kappa \leq 1, c$ | $cW(t-1)$ | $\kappa W(t-1)$ | satisfy (4) |
| [9] LORD'17 | $\sum_{i=1}^{\infty} \gamma_i = 1$ | $\phi_t$ | $\gamma_t W_0 + B_0 \sum_{j:\tau_j < t} \gamma_{t-\tau_j}$ | $B_0 = \alpha - W_0$ |

Table 1: Examples of GAI rules.

### 3.1 Improved monotone GAI rules (GAI++) under independence

In their initial work on GAI rules, Aharoni and Rosset [1] did not incorporate an explicit parameter $B_0$; rather, they proved that choosing $W_0 = B_0 = \alpha$ suffices for $\text{mFDR}_1$ control. In subsequent work, Javanmard and Montanari [9] introduced the parameter $B_0$ and proved that for monotone GAI rules, the same choice $W_0 = B_0 = \alpha$ suffices for $\text{sFDR}_1$ control, whereas the choice $B_0 = \alpha - W_0$ suffices for FDR control, with both results holding under independence. In fact, their monotone GAI rules with $B_0 = \alpha - W_0$ are the only known methods that control FDR. This state of affairs leads to the following dilemma raised in their paper [9]:

> A natural question is whether, in practice, we should choose $W_0, B_0$ as to guarantee FDR control (and hence set $B_0 = \alpha - W_0 \ll \alpha$) or instead be satisfied with mFDR or sFDR control, which allow for $B_0 = \alpha$ and hence potentially larger statistical power.

Our first contribution is a "win-win" resolution to this dilemma: more precisely, we prove that we can choose $B_0 = \alpha$ while maintaining FDR control, with a small catch that at the *very first rejection only*, we need $B_0 = \alpha - W_0$. Of course, in this case $B_0$ is not constant, and hence we replace it by the random variable $b_t \in \mathcal{F}^{t-1}$, and we prove that choosing $W_0, b_t$ such that $b_t + W_0 = \alpha$ for the first rejection, and simply $b_t = \alpha$ for every future rejection, suffices for formally proving FDR control under independence. This achieves the best of both worlds (guaranteeing FDR control, and handing out the largest possible reward of $\alpha$), as posed by the above dilemma. To restate our contribution, we effectively prove that the power of monotone GAI rules can be uniformly improved without changing the FDR guarantee.

Formally, we define our improved generalized alpha-investing (GAI++) algorithm as follows. It sets $W(0) = W_0$ with $0 \le W_0 \le \alpha$, and chooses $\alpha_t \in \mathcal{F}^{t-1}$ to make decisions $R_t = \mathbf{1}\{P_t \le \alpha_t\}$ and updates the wealth $W(t) = W(t-1) - \phi_t + R_t \psi_t \in \mathcal{F}^t$ using some $\phi_t \le W(t-1) \in \mathcal{F}^{t-1}$ and some reward $\psi_t \le \min\{\phi_t + b_t, \frac{\phi_t}{\alpha_t} + b_t - 1\} \in \mathcal{F}^{t-1}$, using the choice

$$b_t = \begin{cases} \alpha - W_0 & \text{when} \quad R(t-1) = 0 \\ \alpha & \text{otherwise} \end{cases} \quad \in \mathcal{F}^{t-1}.$$

As an explicit example, given an infinite nonincreasing sequence of positive constants $\{\gamma_j\}$ that sums to one, the LORD++ algorithm effectively makes the choice:

$$\alpha_t = \gamma_t W_0 + (\alpha - W_0)\gamma_{t-\tau_1} + \alpha \sum_{j:\tau_j < t, \tau_j \ne \tau_1} \gamma_{t-\tau_j}, \tag{5}$$

recalling that $\tau_j$ is the time of the $j$-th rejection. Reasonable default choices include $W_0 = \alpha/2$, and $\gamma_j = 0.0722 \frac{\log(j \vee 2)}{j e^{\sqrt{\log j}}}$, the latter derived in the context of testing if a Gaussian is zero mean [9].

Any monotone GAI++ rule comes with the following guarantee.

**Theorem 1.** *Any monotone GAI++ rule satisfies the bound* $\mathbb{E}\left[\frac{V(T) + W(T)}{R(T)}\right] \le \alpha$ *for all* $T \in \mathbb{N}$ *under independence. Since* $W(T) \ge 0$ *for all* $T \in \mathbb{N}$*, any such rule (a) controls FDR at level* $\alpha$ *under independence, and (b) has power at least as large as the corresponding GAI algorithm.*

The proof of this theorem is provided in Appendix F. Note that for monotone rules, a larger alpha-wealth reward at each rejection yields a possibly higher power, but never lower power, immediately implying statement (b). Consequently, we provide only a proof for statement (a) in Appendix F. For the reader interested in technical details, a key super-uniformity Lemma 1 and associated intuition for online FDR algorithms is provided in Section 3.2.

## 3.2 Intuition for larger rewards via a super-uniformity lemma

For the purposes of providing some intuition for why we are able to obtain larger rewards than Javanmard and Montanari [9], we present the following lemma. In order to set things up, recall that $R_t = \mathbf{1}\{P_t \le \alpha_t\}$ and note that $\alpha_t$ is $\mathcal{F}^{t-1}$-measurable, being a coordinatewise nondecreasing function of $R_1, \ldots, R_{t-1}$. Hence, the marginal super-uniformity assumption (1) immediately implies that for independent $p$-values, we have

$$\Pr\{P_t \le \alpha_t \mid \mathcal{F}^{t-1}\} \le \alpha_t, \quad \text{or equivalently,} \quad \mathbb{E}\left[\frac{\mathbf{1}\{P_t \le \alpha_t\}}{\alpha_t} \mid \mathcal{F}^{t-1}\right] \le 1. \tag{6}$$

Lemma 1 states that under independence, the above statement remains valid in much more generality.

Given a sequence $P_1, P_2, \ldots$ of independent $p$-values, define a filtration via the sigma-fields $\mathcal{F}^{i-1} := \sigma(R_1, \ldots, R_{i-1})$, where $R_i := \mathbf{1}\{P_i \le f_i(R_1, \ldots, R_{i-1})\}$ for some coordinatewise nondecreasing function $f_i : \{0,1\}^{i-1} \to \mathbb{R}$. With this set-up, we have the following guarantee:

**Lemma 1.** *Let* $g : \{0,1\}^T \to \mathbb{R}$ *be any coordinatewise nondecreasing function such that* $g(\vec{x}) > 0$ *for any vector* $\vec{x} \ne (0, \ldots, 0)$*. Then for any index* $t \le T$ *such that* $H_t \in \mathcal{H}^0$*, we have*

$$\mathbb{E}\left[\frac{\mathbf{1}\{P_t \le f_t(R_1, \ldots, R_{t-1})\}}{g(R_1, \ldots, R_T)} \mid \mathcal{F}^{t-1}\right] \le \mathbb{E}\left[\frac{f_t(R_1, \ldots, R_{t-1})}{g(R_1, \ldots, R_T)} \mid \mathcal{F}^{t-1}\right]. \tag{7}$$

This super-uniformity lemma is analogous to others used in offline multiple testing [4, 11], and will be needed in its full generality later in the paper. The proof of this lemma in Appendix E is based on a leave-one-out technique which is common in the multiple testing literature [7, 10, 11]; ours specifically generalizes a lemma in the Appendix of Javanmard and Montanari [9].

As mentioned, this lemma helps to provide some intuition for the condition on $\psi_t$ and the unorthodox condition on $b_t$. Indeed, note that by definition,

$$\text{FDR}(T) = \mathbb{E}\left[\frac{V(T)}{R(T)}\right] = \mathbb{E}\left[\frac{\sum_{t\in\mathcal{H}^0}\mathbf{1}\left\{P_t\le\alpha_t\right\}}{R(T)}\right] \le \mathbb{E}\left[\frac{\sum_{t=1}^T\alpha_t}{\sum_{t=1}^T R_t}\right],$$

where we applied Lemma 1 to the coordinatewise nondecreasing function $g(R_1,\ldots,R_T) = R(T)$. From this equation, we may infer the following: If $\sum_t R_t = k$, then the FDR will be bounded by $\alpha$ as long as the total alpha-wealth $\sum_t \alpha_t$ that was used for testing is smaller than $k\alpha$. In other words, with every additional rejection that adds one to the denominator, the algorithm is allowed extra alpha-wealth equaling $\alpha$ for testing.

In order to see where this shows up in the algorithm design, assume for a moment that we choose our penalty as $\phi_t = \alpha_t$. Then, our condition on rewards $\psi_t$ simply reduces to $\psi_t \le b_t$. Furthermore, since we choose $b_t = \alpha$ after every rejection except the first, our total *earned* alpha-wealth is approximately $\alpha R(T)$, which also upper bounds the total alpha-wealth used for testing.

The intuitive reason that $b_t$ cannot equal $\alpha$ at the very first rejection can also be inferred from the above equation. Indeed, note that because of the definition of FDR, we have $\frac{V(T)}{R(T)} := \frac{V(T)}{R(T)\vee 1}$, the denominator $R(T)\vee 1 = 1$ when the number of rejections equals zero or one. Therefore, the denominator only starts incrementing at the second rejection. Hence, the sum of $W_0$ and the first reward must be at most $\alpha$, following which one may award $\alpha$ at every rejection. This is the central piece of intuition behind the GAI algorithm design, its improvement in this paper, and the FDR control analysis. To the best of our knowledge, this is the first explicit presentation for the intuition for online FDR control.

## 4  A direct method for deriving new online FDR rules

Many offline FDR procedures can be derived in terms of an estimate $\widehat{\text{FDP}}$ of the false discovery proportion; see Ramdas et al. [11] and references therein. The discussion in Section 3.2 suggests that it is also possible to write online FDR rules in this fashion. Indeed, given any non-negative, predictable sequence $\{\alpha_t\}$, we propose the following definition:

$$\widehat{\text{FDP}}(t) := \frac{\sum_{j=1}^t \alpha_j}{R(t)}.$$

This definition is intuitive because $\widehat{\text{FDP}}(t)$ approximately overestimates the unknown $\text{FDP}(t)$:

$$\widehat{\text{FDP}}(t) \ge \frac{\sum_{j\le t, j\in\mathcal{H}^0}\alpha_j}{R(t)} \approx \frac{\sum_{j\le t, j\in\mathcal{H}^0}\mathbf{1}\left\{P_j\le\alpha_j\right\}}{R(t)} = \text{FDP}(t).$$

A more direct way to construct new online FDR procedures is to ensure that $\sup_{t\in\mathbb{N}}\widehat{\text{FDP}}(t) \le \alpha$, bypassing the use of wealth, penalties and rewards in GAI. This idea is formalized below.

**Theorem 2.** *For any predictable sequence $\{\alpha_t\}$ such that $\sup_{t\in\mathbb{N}}\widehat{\text{FDP}}(t) \le \alpha$, we have:*
*(a) If the p-values are super-uniform conditional on all past discoveries, meaning that $Pr\{P_j \le \alpha_j \mid F^{j-1}\} \le \alpha_j$, then the associated procedure has $\sup_{T\in\mathbb{N}}\text{mFDR}(T) \le \alpha$.*
*(b) If the p-values are independent and if $\{\alpha_t\}$ is monotone, then we also have $\sup_{T\in\mathbb{N}}\text{FDR}(T) \le \alpha$.*

The proof of this theorem is given in Appendix D. In our opinion, it is more transparent to verify that LORD++ controls both mFDR and FDR using Theorem 2 than using Theorem 1.

## 5  Incorporating prior and penalty weights

Here, we develop GAI++ algorithms that incorporate *prior weights* $w_t$, which allow the user to exploit domain knowledge about which hypotheses are more likely to be non-null, as well as *penalty weights* $u_t$ to differentiate more important hypotheses from the rest. The weights must be strictly positive, predictable (meaning that $w_t, u_t \in \mathcal{F}_{t-1}$) and monotone (in the sense of definition (3)).

**Penalty weights.** For many motivating applications, including internet companies running a series of A/B tests over time, or drug companies doing a series of clinical trials over time, it is natural to assume that some tests are more important than others, in the sense that some false discoveries may have more lasting positive/negative effects than others. To incorporate this in the offline setting, Benjamini and Hochberg [3] suggested associating each test with a positive *penalty* weight $u_i$ with hypothesis $H_i$. Choosing $u_i > 1$ indicates a more impactful or important test, while $u_i < 1$ means the opposite. Although algorithms exist in the offline setting that can intelligently incorporate penalty weights, no such flexibility currently exists for online FDR algorithms. With this motivation in mind and following Benjamini and Hochberg [3], define the penalty-weighted FDR as

$$\text{FDR}_u(T) := \mathbb{E}\left[\frac{V_u(T)}{R_u(T)}\right] \tag{8}$$

where $V_u(T) := \sum_{t \in \mathcal{H}^0} u_t R_t = V_u(T-1) + u_T R_T \mathbf{1}\{T \in \mathcal{H}^0\}$ and $R_u(T) := R_u(T-1) + u_T R_T$. One may set $u_t = 1$ to recover the special case of no penalty weights. In the offline setting, a given set of penalty weights can be rescaled to make the average penalty weight equal unity, without affecting the associated procedure. However, in the online setting, we choose penalty weights $u_t$ one at a time, possibly not knowing the total number of hypotheses ahead of time. As a consequence, these weights cannot be rescaled in advance to keep their average equal to unity. It is important to note that we allow $u_t \in \mathcal{F}_{t-1}$ to be determined *after* viewing the past rejections, another important difference from the offline setting. Indeed, if the hypotheses are logically related (even if the p-values are independent), then the current hypothesis can be more or less critical depending on which other ones are already rejected.

**Prior weights.** In many applications, one may have access to prior knowledge about the underlying state of nature (that is, whether the hypothesis is truly null or non-null). For example, an older published biological study might have made significant discoveries, or an internet company might know the results of past A/B tests or decisions made by other companies. This knowledge may be incorporated by a weight $w_t$ which indicates the strength of a prior belief about whether the hypothesis is null or not—typically, a larger $w_t > 1$ can be interpreted as a greater likelihood of being a non-null, indicating that the algorithm may be more aggressive in deciding whether to reject $H_t$. Such p-value weighting was first suggested in the offline FDR context by [6], though earlier work employed it in the context of FWER control. As with penalty weights in the offline setting, offline prior weights are also usually rescaled to have unit mean, and then existing offline algorithms simply replace the p-value $P_t$ by the weighted p-value $P_t/w_t$. However, it is not obvious how to incorporate prior weights in the online setting. As we will see in the sections to come, the online FDR algorithms we propose will also use p-value reweighting; moreover, the rewards must be prudently adjusted to accommodate the fact that an a-priori rescaling is not feasible. Furthermore, as opposed to the offline case, the weights $w_t \in \mathcal{F}_{t-1}$ are allowed to depend on past rejections. This additional flexibility allows one to set the weights not only based on our prior knowledge of the current hypothesis being tested, but also based on properties of the sequence of discoveries (for example, whether we recently saw a string of rejections or non-rejections). We point out some practical subtleties with the use and interpretation of prior weights in Appendix C.4.

**Doubly-weighted GAI++ rules.** Given a testing level $\alpha_t$ and weights $w_t, u_t$, all three being predictable and monotone, we make the decision

$$R_t := \mathbf{1}\{P_t \leq \alpha_t u_t w_t\}. \tag{9}$$

This agrees with the intuition that larger prior weights should be reflected in an increased willingness to reject the null, and we should favor rejecting more important hypotheses. As before, our rejection reward strategy differs before and after $\tau_1$, the time of the first rejection. Starting with some $W(0) = W_0 \leq \alpha$, we update the wealth as $W(t) = W(t-1) - \phi_t + R_t \psi_t$, where $w_t, u_t, \alpha_t, \phi_t, \psi_t \in \mathcal{F}^{t-1}$ must be chosen so that $\phi_t \leq W(t-1)$, and the rejection reward $\psi_t$ must obey the condition

$$0 \leq \psi_t \leq \min\left\{\phi_t + u_t b_t, \frac{\phi_t}{u_t w_t \alpha_t} + u_t b_t - u_t\right\}, \quad \text{where} \tag{10a}$$

$$b_t := \alpha - \frac{W_0}{u_t}\mathbf{1}\{\tau_1 > t-1\} \in \mathcal{F}_{t-1}. \tag{10b}$$

Notice that setting $w_t = u_t = 1$ immediately recovers the GAI updates. Let us provide some intuition for the form of the rewards $\psi_t$, which involves an interplay between the weights $w_t, u_t$,

the testing levels $\alpha_t$ and the testing penalties $\phi_t$. First note that large weights $u_t, w_t > 1$ result in a smaller earning of alpha-wealth and if $\alpha_t, \phi_t$ are fixed, then the maximum "common-sense" weights are determined by requiring $\psi_t \geq 0$. The requirements of lower rewards for larger weights and of a maximum allowable weight should both seem natural; indeed, there must be some price one must pay for an easier rejection, otherwise we would always use a high prior weight or penalty weight to get more power, no matter the hypothesis! We show that such a price does not have to be paid in terms of the FDR guarantee—we prove that $\mathrm{FDR}_u$ is controlled for any choices of weights—but a price is paid in terms of power, specifically the ability to make rejections in the future. Indeed, the combined use of $u_t, w_t$ in both the decision rule $R_t$ and the earned reward $\psi_t$ keeps us honest; if we overstate our prior belief in the hypothesis being non-null or its importance by assigning a large $u_t, w_t > 1$, we will not earn much of a reward (or even a negative reward!), while if we understate our prior beliefs by assigining a small $u_t, w_t < 1$, then we may not reject this hypothesis. Hence, it is prudent to not misuse or overuse the weights, and we recommend that the scientist uses the default $u_t = w_t = 1$ in practice unless there truly is prior evidence against the null or a reason to believe the finding would be of importance, perhaps due to past studies by other groups or companies, logical relationships between hypotheses, or due to extraneous reasons suggested by the underlying science.

We are now ready to state a theoretical guarantee for the doubly-weighted GAI++ procedure:

**Theorem 3.** *Under independence, the doubly-weighted GAI++ algorithm satisfies the bound* $\mathbb{E}\left[\frac{V_u(T) + W(T)}{R_u(T)}\right] \leq \alpha$ for all $T \in \mathbb{N}$. *Since* $W(T) \geq 0$, *we also have* $\mathrm{FDR}_u(T) \leq \alpha$ *for all* $T \in \mathbb{N}$.

The proof of this theorem is given in Appendix G. It is important to note that although we provide the proof here only for GAI++ rules under independence, the ideas would actually carry forward in an analogous fashion for GAI rules under various other forms of dependence.

## 6 From infinite to decaying memory

Here, we summarize two phenomena : (i) the "piggybacking" problem that can occur with non-stationary null-proportion, (ii) the "alpha-death" problem that can occur with a sequence of nulls. We propose a new error metric, the decaying-memory FDR (mem-FDR), that for truly temporal multiple testing scenarios, and propose an adjustment of our GAI++ algorithms to control this quantity.

**Piggybacking.** As outlined in motivation M1, when the full batch of $p$-values is available offline, online FDR algorithms have an inherent asymmetry in their treatment of different $p$-values, and make different rejections depending on the order in which they process the batch. Indeed, Foster and Stine [5] demonstrated that if one knew a reasonably good ordering (with non-nulls arriving earlier), then their online alpha-investing procedures could attain higher power than the offline BH procedure. This is partly due to a phenomenon that we call "piggybacking"—if a lot of rejections are made early, these algorithms earn and accumulate enough alpha-wealth to reject later hypotheses more easily by testing them at more lenient thresholds than earlier ones. In essence, later tests "piggyback" on the success of earlier tests. While piggybacking may be desirable or acceptable under motivation M1, such behavior may be unwarranted and unwanted under motivation M2. We argue that piggybacking may lead to a spike in the false discovery rate *locally in time*, even though the FDR over all time is controlled. This may occur when the sequence of hypotheses is non-stationary and clustered, when strings of nulls may follow strings of non-nulls. For concreteness, consider the setting in Javanmard and Montanari [8] where an internet company conducts many A/B tests over time. In "good times", when a large fraction tests are truly non-null, the company may accumulate wealth due to frequent rejections. We demonstrate using simulations that such accumulated wealth can lead to a string of false discoveries when there is a quick transition to a "bad period" where the proportion of non-nulls is much lower, causing a spike in the false discovery proportion *locally in time*.

**Alpha-death.** Suppose we test a long stretch of nulls, followed by a stretch of non-nulls. In this setting, GAI algorithms will make (almost) no rejections in the first stretch, losing nearly all of its wealth. Thereafter, the algorithm may be effectively condemned to have no power, unless a non-null with extremely strong signal is observed. Such a situation, from which no recovery is possible, is perfectly reasonable under motivation M1. The alpha-wealth has been used up fully, and those are the only rejections we are allowed to make with that batch of $p$-values. However, for an internet company operating with motivation M2, it might be unacceptable to inform them that they essentially cannot run any more tests, or that they may perhaps never make another useful discovery.

Both of these problems, demonstrated in simulations in Appendix C.2, are due to the fact that the process effectively has an infinite memory. In the following, we propose one way to smoothly forget the past and to some extent alleviate the negative effects of the aforementioned phenomena.

**Decaying memory.** For a user-defined decay parameter $\delta > 0$, define $V^\delta(0) = R^\delta(0) = 0$ and define the *decaying memory FDR* as follows:

$$\text{mem-FDR}(T) := \mathbb{E}\left[\frac{V^\delta(T)}{R^\delta(T)}\right],$$

where $V^\delta(T) := \delta V^\delta(T-1) + R_T \mathbf{1}\left\{T \in \mathcal{H}^0\right\} = \sum_{t \in \mathcal{H}^0} \delta^{T-t} R_t \mathbf{1}\left\{t \in \mathcal{H}^0\right\}$, and analogously $R^\delta(T) := \delta R^\delta(T-1) + R_T = \sum_t \delta^{T-t} R_t$. This notion of FDR control, which is arguably natural for modern temporal applications, appears to be novel in the multiple testing literature. The parameter $\delta$ is reminiscent of the discount factor in reinforcement learning.

**Penalty-weighted decaying-memory FDR.** We may naturally extend the notion of decaying-memory FDR to encompass penalty weights. Setting $V_u^\delta(0) = R_u^\delta(0) = 0$, we define

$$\text{mem-FDR}_u(T) := \mathbb{E}\left[\frac{V_u^\delta(T)}{R_u^\delta(T)}\right],$$

where we define $V_u^\delta(T) := \delta V_u^\delta(T-1) + u_T R_T \mathbf{1}\left\{T \in \mathcal{H}^0\right\} = \sum_{t=1}^T \delta^{T-t} u_t R_t \mathbf{1}\left\{t \in \mathcal{H}^0\right\}$, $R_u^\delta(T) := \delta R_u^\delta(T-1) + u_t R_t = \sum_{t=1}^T \delta^{T-t} u_t R_t$.

**mem-GAI++ algorithms with decaying memory and weights.** Given a testing level $\alpha_t$, we make the decision using equation (9) as before, starting with a wealth of $W(0) = W_0 \leq \alpha$. Also, recall that $\tau_k$ is the time of the $k$-th rejection. On making the decision $R_t$, we update the wealth as:

$$W(t) := \delta W(t-1) + (1-\delta)W_0 \mathbf{1}\left\{\tau_1 > t-1\right\} - \phi_t + R_t \psi_t, \tag{11}$$

$$\text{so that} \quad W(T) = W_0 \delta^{T-\min\{\tau_1, T\}} + \sum_{t=1}^T \delta^{T-t}(-\phi_t + R_t \psi_t).$$

The first term in equation (11) indicates that the wealth must decay in order to forget the old earnings from rejections far in the past. If we were to keep the first term and drop the second, then the effect of the initial wealth (not just the post-rejection earnings) also decays to zero. Intuitively, the correction from the second term suggests that even if one forgets all the past post-rejection earnings, the algorithm should behave as if it started from scratch, which means that its initial wealth should not decay. This does not contradict the fact that initial wealth can be *consumed* because of testing penalties $\phi_t$, but it should not *decay* with time—the decay was only introduced to avoid piggybacking, which is an effect of post-rejection *earnings* and not the *initial* wealth.

A natural restriction on $\phi_t$ is the bound $\phi_t \leq \delta W(t-1) + (1-\delta)W_0 \mathbf{1}\left\{\tau_1 > t-1\right\}$, which ensures that the wealth stays non-negative. Further, $w_t, u_t, \alpha_t, \phi_t \in \mathcal{F}^{t-1}$ must be chosen so that the rejection reward $\psi_t$ obeys conditions (10a) and (10b). Notice that setting $w_t = u_t = \delta = 1$ recovers the GAI++ updates. As an example, mem-LORD++ would use :

$$\alpha_t = \gamma_t W_0 \delta^{t-\min\{\tau_1, t\}} + \sum_{j:\tau_j < t} \delta^{t-\tau_j} \gamma_{t-\tau_j} \psi_{\tau_j}.$$

We are now ready to present our last main result.

**Theorem 4.** *Under independence, the doubly-weighted mem-GAI++ algorithm satisfies the bound* $\mathbb{E}\left[\frac{V_u^\delta(T) + W(T)}{R_u^\delta(T)}\right] \leq \alpha$ *for all $T \in \mathbb{N}$. Since $W(T) \geq 0$, we have* $\text{mem-FDR}_u(T) \leq \alpha$ *for all $T \in \mathbb{N}$.*

See Appendix H for the proof of this claim. Appendix B discusses how to use "abstaining" to provide a smooth restart from alpha-death, whereas Appendix C contains a numerical simulation demonstrating the use of decaying memory.

## 7  Summary

In this paper, we make four main contributions—more powerful procedures under independence, an alternate viewpoint of deriving online FDR procedures, incorporation of prior and penalty weights, and introduction of a decaying-memory false discovery rate to handle piggybacking and alpha-death. Numerical simulations in Appendix C complement the theoretical results.

**Acknowledgments**

We thank A. Javanmard, R. F. Barber, K. Johnson, E. Katsevich, W. Fithian and L. Lei for related discussions, and A. Javanmard for sharing code to reproduce experiments in Javanmard and Montanari [9]. This material is based upon work supported in part by the Army Research Office under grant number W911NF-17-1-0304, and National Science Foundation grant NSF-DMS-1612948.

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
