[Supplementary Material · online-nips-app.pdf]

# A  Summary of joint dependence assumptions in previous work

We use the phrase "FDR$_*$ control" to mean the control of either FDR or mFDR or sFDR. It is important to discuss the assumptions on the joint dependence on $p$-values, under which FDR$_*$ control can be proved. These are listed from (approximately) weakest to strongest below:

1. **Arbitrary Dependence.** Null $p$-values are arbitrarily dependent on all other $p$-values.
2. **SuperCoRD.** Null $p$-values are super-uniform conditional on the time of most recent discovery, meaning that for all $t \in \mathcal{H}^0$ and for any $\alpha_t \in \mathcal{F}^{t-1}$, we have
$$\Pr\{P_t \leq \alpha_t \mid \tau_{\mathrm{prev}}\} \leq \alpha_t,$$
where $\tau_{\mathrm{prev}} = \max_{s<t}\{s : R_s = 1\}$ is the time of the previous rejection.
3. **SuperCoND.** Null $p$-values are super-uniform conditional on the number of discoveries up to that point, meaning that for all $t \in \mathcal{H}^0$ and for any $\alpha_t \in \mathcal{F}^{t-1}$, we have
$$\Pr\{P_t \leq \alpha_t \mid R(T-1)\} \leq \alpha_t.$$
4. **SuperCoAD.** Null $p$-values are super-uniform conditional on all discoveries, meaning that for all $t \in \mathcal{H}^0$ and for any $\alpha_t \in \mathcal{F}^{t-1}$, we have
$$\Pr\big\{P_t \leq \alpha_t \mid \mathcal{F}^{t-1}\big\} \leq \alpha_t.$$
5. **Independence.** Null $p$-values are independent of all other $p$-values.

Table 2 summarizes some known algorithms, the dependence these algorithms can handle, and the type of FDR control they guarantee. Of special note is an algorithm called LORD [9] that the authors noted performs consistently well in practice, and thus will be the focus of most of our experiments (the conclusions of which carry forward qualitatively to other monotone algorithms).

| Ref. | Algorithm | Dependence | Control (at any $T$) | Monotone? |
|---|---|---|---|---|
| - | Alpha-spending | Arbitrary | FWER$(T)$ | No |
| [5] | Alpha-investing (AI) | SuperCoAD | mFDR$_\eta(T)$ | No |
| [1] | Generalized Alpha-investing (GAI) | SuperCoAD | mFDR$_\eta(T)$ | No |
| [8] | Levels based on Number of Discoveries (LOND) | SuperCoND | mFDR$_\eta(T)$, FDR$(T)$ | Yes |
| [8] | LOND (with a conservative correction) | Arbitrary | FDR$(T)$ | Yes |
| [8] | Levels based on most Recent Disc. (LORD'15) | SuperCoRD | mFDR$_\eta(T)$ | Yes |
| [9] | Monotone GAI (including LORD'17) | Independence | FDR$(T)$, sFDR$_\eta(T)$ | Yes |

Table 2: Summary of previous work. Note that LORD'17 is an improvement over LORD'15 with higher power, and the shorthand "LORD" will be reserved for LORD'17.

# B  Abstinence for recovery from alpha-death

For truly temporal applications as outlined in motivation M2, we allow the algorithm to *abstain* from testing, meaning that it does not need to perform a test at each time step. In this case, we use the convention of $P_t = -1$ to indicate that we abstained from testing at time $t$. Also, we introduce the random variables
$$A_t := \mathbf{1}\{P_t = -1\}, \quad \text{and} \quad A_t^c := 1 - A_t, \tag{12}$$
as indicators for abstention. Abstention may happen due to the natural variation in frequency of testing hypotheses in real-world applications. Additionally, abstention is the natural treatment for recovery from alpha-death. If the alpha-wealth is deemed too low, abstaining for a while can drop mem-FDR below a threshold, and when it becomes small enough, one can reset all variables and restart the entire process. In more detail, note that we would change the quantities $V(t), W(t), R(t)$ only if we actually did not abstain and performed a test, as given by:

$$W(t) := \delta W(t-1) + (1-\delta)W_0 \mathbf{1}\{\tau_1 > t-1\} - A_t^c \phi_t + A_t^c R_t \psi_t$$
$$V_u^\delta(t) := \delta V_u^\delta(t-1) + u_t A_t^c R_t \mathbf{1}\{t \in \mathcal{H}^0\}$$
$$R_u^\delta(t) := \delta R_u^\delta(t-1) + u_t A_t^c R_t.$$

When we abstain, assuming that we have made at least one rejection, all three quantities decay with time. Hence, the ratio $\frac{V_u^\delta(t)+W(t)}{R_u^\delta(t)}$ remains unchanged initially, and when the denominator $R_u^\delta(t)$ falls below one, the aforementioned ratio decays smoothly to zero (and hence so does the

mem-FDR$_u$). Using a user-defined tolerance $\epsilon$, we can then "reset" when $R_u^\delta(t) < \epsilon$ by re-defining all quantities to their starting values, setting the time to zero, and restarting the entire process.

An alternative to abstinence is to pre-define a period of time after which the process will reset, like a calendar year, or a single financial quarter. With this choice, decaying memory may help with piggybacking but is not needed for recovery from alpha-death. However, for applications in which there is no natural special period, and which is in some sense continuous in time without discrete breakpoints, the decaying memory FDR is a natural quantity to control, and abstinence is an arguably intuitive solution to alpha-death. Indeed, companies are obviously less willing to accept a permanent alpha-death that ends all testing forever, and are more likely to be willing to abstain from testing for a while, and run an internal check on why they lost alpha-wealth by testing too many nulls, or perhaps why they had very low signal on their non-nulls (making them seem like nulls).

## C   Numerical Simulations

Here, we provide proof-of-concept experiments for various aspects of the paper.[1]

### C.1   Evidence of higher power of GAI++ over GAI

To demonstrate an improvement of GAI++ over GAI, we follow the simple experimental setup of Javanmard and Montanari [9] which tests the means of $T = 1000$ Gaussian distributions. The null hypothesis is $H_j : \mu_j = 0$ for $j = \{1, \ldots, T\}$. We observe independent $Z_j \sim N(\mu_j, 1)$, which can be converted using the Gaussian CDF $\Phi$ to a one-sided $p$-value, $P_j = \Phi(-Z_j)$ or to a two-sided $p$-value, $P_j = 2\Phi(-|Z_j|)$. Notice that the $p$-value is exactly uniformly distributed when the null hypothesis is true, that is $\mu_j = 0$. The means $\mu_j$ are set independently according to the mixture:

$$\mu_j \sim \begin{cases} 0 \text{ with probability } 1 - \pi_1, \\ N(0, \sigma^2) \text{ with probability } \pi_1, \end{cases}$$

and we set $\sigma^2 = 2 \log T$, resulting in means that are near the boundary of detectability. Indeed, under the global null where $\pi_1 = 0$, $\max_j Z_j = (1 + o(1))\sqrt{2 \log T}$, and $\sqrt{2 \log T}$ is the minimax amplitude for estimation under the sparse Gaussian sequence model.

The improvement in power of GAI++ over GAI depends on the choice of $W_0$ and $B_0 = \alpha - W_0$. If $W_0$ is too small, the algorithm may suffer from alpha-death too quickly, because the signals may not be strong enough for the algorithm to recover by accumulating the large rewards $B_0$. If $W_0$ is too large, the reward $B_0$ at each step will be too small, and the algorithm will suffer from lower power. Hence, the larger $W_0$ is, the smaller $B_0$ is, and the more GAI++ will improve over GAI. For our simulations, we set $W_0 = \alpha/5$, for which we only expect a small improvement, and always have $\alpha = 0.05$, and run 200 independent trials to estimate the power and FDR.

Figure 1: Plots of power vs $\pi_1$ (left panel) and FDR versus $\pi_1$ (right panel, for various algorithms.

For concrete monotone GAI and GAI++ procedures, we choose LORD'17 as detailed in Table 1 and LORD++ from definition (5). We define power as usual in the FDR literature:

$$\text{power}(T) := \mathbb{E}\left[\frac{\sum_{t \in \mathcal{H}^1} R_t}{\sum_{t=1}^{T} \mathbf{1}\{t \in \mathcal{H}^1\}}\right].$$

In Figure 1, we plot the power and FDR for the Bonferroni and LORD algorithms using $W_0 = \alpha/2$ and the constant sequenuce $\gamma_j = 0.0722 \frac{\log(j \vee 2)}{j e^{\sqrt{\log j}}}$ derived for testing Gaussian means [9], where the leading constant was approximated so that the infinite sequence sums to one. As predicted by the theory, the power of the LORD++ algorithm is uniformly better than LORD.

## C.2  Piggybacking and decaying memory

For this subsection, we move away from the stationary setting that is a useful base case, but unrealistic in practice. To bring out the phenomenon of piggybacking, we consider the setting where $\pi_1 \gg 0.5$ in the first 1000 tests, and $\pi_1 \ll 0.5$ in the second 1000. There is nothing specific to this particular choice, and will qualitatively occur whenever there is a stretch of non-nulls followed by a stretch of nulls. For simplicity, we restrict our attention to the LORD++ and the mem-LORD++ algorithms, and plot their mem-FDR as a function of time. In particular, we use the following concrete update for $\alpha_t$ in the mem-LORD++ algorithm:

$$\alpha_t = \gamma_t W_0 \delta^{t-\min\{\tau_1, t\}} + (\alpha - W_0)\delta^{t-\tau_1}\gamma_{t-\tau_1} + \alpha\Big(\sum_{\tau_j < t, \tau_j \neq \tau_1} \delta^{t-\tau_j}\gamma_{t-\tau_j}\Big).$$

Figure 2: Plots of mem-power versus time (left panel and mem-FDR versus time (right panel), for LORD++ and mem-LORD++ with $\delta = 0.99$. The spike in false discoveries suffered by LORD++ due to piggybacking is significantly smoothed by mem-LORD++ without much loss of power.

Figure 2 demonstrates see that LORD++ suffers a large spike in mem-FDR locally in time, which is significantly smoothed out by mem-LORD++ with $\delta = 0.99$, at an insignificant loss of power. Arguably, the power should itself be replaced by a "decaying memory power" which we call mem-power, which definition is analogous to mem-FDR in relation to FDR, i.e.

$$\text{mem-power}(T) := \mathbb{E}\left[\frac{U^\delta(T)}{D^\delta(T)}\right],$$

where $U^\delta(T) := \delta U^\delta(T-1) + R_T \mathbf{1}\{T \in \mathcal{H}^1\}$ and $D^\delta(T) := \delta D^\delta(T-1) + \mathbf{1}\{T \in \mathcal{H}^1\}$. Due to its conservative choice for $\alpha_t$, the smoothing of the mem-FDR measure comes at the expense of lower mem-power for mem-LORD++ compared to LORD++ in the second half of the experiment.

## C.3  Alpha-death

In this section we illustrate the usefulness of abstinence as discussed in Section B for experiments where alpha-death is reached rather quickly. Concretely, we choose the probability of each hypothesis being non-null to be identically and independently $p = 0.01$. Furthermore, we abstain from testing if $W(t) < \epsilon_w$ and we reset to initial values if $R(t) < \epsilon_r$ with $\epsilon_w = 0.05W_0$ and $\epsilon_r = 0.1$.

Figure 3: Plots of wealth versus time (left panel) power versus time (right panel), for mem-LORD++ with $\delta = 0.99$ and a constant $\pi_1 = 0.01$. Once the wealth vanishes, the generic mem-LORD++ cannot make new discoveries for the entire future, whereas the abstinent mem-LORD++ circumvents this issue and eventually starts anew, allowing new incoming non-nulls to be detected.

Figure 3 depicts both the time development of wealth on the left hand side and the corresponding mem-power on the right hand side. The red curves representing the generic mem-LORD++ algorithm show that once wealth reaches 0, no discoveries can be made so that mem-power stays at 0 for the entire rest of the experiment. On the other hand, for the exact same experiment, the abstinent mem-LORD++ in green has a "second chance" after abstaining for a while: the experiment is reset so that new discoveries can be made even though the wealth had depleted at some previous time.

## C.4  Subtleties with the use of prior weights

If one has a high prior belief that a hypothesis is non-null, then the "oracle" strategy of assigning weights depends on the strength of the underlying signal: (a) if the signal is small, an oracle would assign a weight that is just high enough to reject the non-null, while earning a small reward, and (b) if the signal is large, then an oracle would assign a weight as small as possible to just reject the non-null, earning as large a reward as possible, amassing alpha-wealth to be used for later tests.

Figure 4 suggests that in the aforementioned simulation setup, we happened to be in situation (b), where most non-nulls had enough signal so that using a weight smaller than one was more beneficial than a weight larger than one. We used the same setup as the previous subsection, except that we assign "oracle" weights of $1+a$ whenever the hypothesis is non-null, and a weight of $1-a$ whenever the hypothesis is null, for positive and negative choices of $a$. We use the word "oracle" since, in practice, we of course do not know which hypotheses are null and non-null.

Figure 4: Plots of power vs $\pi_1$ (left panel) and FDR versus $\pi_1$ (right panel), for LORD++ with weights $1 + a$ on non-nulls and $1 - a$ on nulls.

## D    Proof of Theorem 2

For any time $T \in \mathbb{N}$, we may infer that mFDR is controlled using the following argument :

$$\mathbb{E}\left[V(T)\right] = \sum_{j \in \mathcal{H}^0, j \leq T} \mathbb{E}\left[\mathbb{E}\left[\mathbf{1}\left\{P_j \leq \alpha_j\right\} \mid \mathcal{F}^{j-1}\right]\right]$$

$$\overset{(i)}{\leq} \sum_{j \in \mathcal{H}^0, j \leq T} \mathbb{E}\left[\alpha_j\right]$$

$$\overset{(ii)}{\leq} \mathbb{E}\left[\sum_{j \leq T} \alpha_j\right]$$

$$\overset{(iii)}{\leq} \alpha \mathbb{E}\left[R(T) \vee 1\right],$$

where inequality $(i)$ follows after taking iterated expectations by conditioning on $\mathcal{F}^{j-1}$, and then applying the conditional superuniformity property (6), inequality $(ii)$ follows simply by dropping the condition $j \in \mathcal{H}^0$, and inequality $(iii)$ follows by the theorem assumption that $\frac{\sum_{j \leq T} \alpha_j}{R(T)} \leq \alpha$. Rearranging yields the conclusion $\text{mFDR}(T) = \frac{\mathbb{E}[V(T)]}{\mathbb{E}[R(T)]} \leq \alpha$, as desired.

When the sequence $\{\alpha_t\}$ is additionally monotone, we can use the following argument to prove that the procedure controls FDR at any time $T \in \mathbb{N}$ :

$$\text{FDR} = \mathbb{E}\left[\frac{V(T)}{R(T)}\right] = \sum_{j \in \mathcal{H}^0, j \leq T} \mathbb{E}\left[\frac{\mathbf{1}\left\{P_j \leq \alpha_j\right\}}{R(T)}\right]$$

$$\overset{(iv)}{\leq} \sum_{j \in \mathcal{H}^0, j \leq T} \mathbb{E}\left[\frac{\alpha_j}{R(T)}\right]$$

$$\overset{(v)}{\leq} \mathbb{E}\left[\frac{\sum_{j \leq T} \alpha_j}{R(T)}\right]$$

$$\overset{(vi)}{\leq} \alpha,$$

where inequality $(iv)$ follows after taking iterated expectations by conditioning on $\mathcal{F}^{j-1}$, and then applying the conditional superuniformity lemma Lemma 1, and inequalities $(v)$ and $(vi)$ follow for the same reasons as inequalities $(ii)$ and $(iii)$.

This concludes the proof of both parts of the theorem.

## E    Proof of Lemma 1

Letting $\vec{P} = (P_1, \ldots, P_T)$ be the original vector of $p$-values, we define a "hallucinated" vector of $p$-values $\widetilde{P}^{-t} := (\widetilde{P}_1, \ldots, \widetilde{P}_T)$ that equals $\vec{P}$, except that the $t$-th component is set to zero :

$$\widetilde{P}_i = \begin{cases} 0 & \text{if } i = t \\ P_i & \text{if } i \neq t. \end{cases}$$

For all $i$, define $\widetilde{R}_i = \mathbf{1}\left\{\widetilde{P}_i \leq f_i(\widetilde{R}_1, \ldots, \widetilde{R}_{i-1})\right\}$ and let the corresponding vectors of rejections using $\vec{P}$ and $\widetilde{P}^{-t}$ be $\vec{R} = (R_1, \ldots, R_T)$ and $\widetilde{R}^{-t} = (\widetilde{R}_1, \ldots, \widetilde{R}_T)$. By construction, we have $\widetilde{R}_i = R_i$ for all $i < t$, and $\widetilde{R}_i \geq R_i$ for all $i \geq t$, from which we conclude that $f_i(R_1, \ldots, R_{i-1}) = f_i(\widetilde{R}_1, \ldots, \widetilde{R}_{i-1})$ for all $i \leq t$. Also, we know $\widetilde{R}_t = 1$ by construction since $\widetilde{P}_t = 0$ implying that $g(\widetilde{R}^{-t}) > 0$. Hence, on the event $\{P_t \leq f_t(R_1, \ldots, R_{t-1})\}$, we have $R_t = \widetilde{R}_t = 1$ and hence also $\vec{R} = \widetilde{R}^{-t}$. This allows us to conclude that

$$\frac{\mathbf{1}\left\{P_t \leq f_t(R_1, \ldots, R_{t-1})\right\}}{g(\vec{R})} = \frac{\mathbf{1}\left\{P_t \leq f_t(R_1, \ldots, R_{t-1})\right\}}{g(\widetilde{R}^{-t})}.$$

Since $\widetilde{R}^{-t}$ is independent of $P_t$, we may take conditional expectations to obtain

$$\mathbb{E}\left[\frac{\mathbf{1}\{P_t \le f_t(R_1,\ldots,R_{t-1})\}}{g(\vec{R})} \;\middle|\; \mathcal{F}^{t-1}\right] = \mathbb{E}\left[\frac{\mathbf{1}\{P_t \le f_t(R_1,\ldots,R_{t-1})\}}{g(\widetilde{R}^{-t})} \;\middle|\; \mathcal{F}^{t-1}\right]$$

$$\overset{(i)}{\le} \mathbb{E}\left[\frac{f_t(R_1,\ldots,R_{t-1})}{g(\widetilde{R}^{-t})} \;\middle|\; \mathcal{F}^{t-1}\right]$$

$$\overset{(ii)}{\le} \mathbb{E}\left[\frac{f_t(R_1,\ldots,R_{t-1})}{g(\vec{R})} \;\middle|\; \mathcal{F}^{t-1}\right],$$

where inequality (i) follows by taking expectation only with respect to $P_t$ by invoking the conditional super-uniformity property (6); and inequality (ii) follows because $g(\vec{R}) \le g(\widetilde{R}^{-t})$ since $R_i \le \widetilde{R}_i$ for all $i$ by monotonicity of the online FDR rule. This concludes the proof of the lemma.

## F  Proof of Theorem 1

Substituting the definitions of $V(T) = \sum_{t=1}^{T} R_t \mathbf{1}\{t \in \mathcal{H}^0\}$ and the alpha-wealth

$$W(T) = W_0 + \sum_{t=1}^{T} (-\phi_t + R_t \psi_t),$$

we may use the tower property of conditional expectation to write

$$\mathbb{E}\left[\frac{V(T) + W(T)}{R(T)}\right] = \sum_t \underbrace{\mathbb{E}\left[\mathbb{E}\left[\frac{\frac{W_0}{T} + R_t(\psi_t + \mathbf{1}\{t \in \mathcal{H}^0\}) - \phi_t}{R(T)} \;\middle|\; \mathcal{F}^{t-1}\right]\right]}_{L_t}.$$

We tackle the above expression term by term, depending on whether or not $t \in \mathcal{H}^0$.

**Case 1.**  First, suppose that $t \in \mathcal{H}^0$. Substituting $\psi_t \le \frac{\phi_t}{\alpha_t} + b_t - 1$ into the expression for $L_t$ yields

$$L_t \le \mathbb{E}\left[\mathbb{E}\left[\frac{\frac{W_0}{T} + R_t(\frac{\phi_t}{\alpha_t} + b_t) - \phi_t}{R(T)} \;\middle|\; \mathcal{F}^{t-1}\right]\right]$$

$$= \mathbb{E}\left[\mathbb{E}\left[\frac{\frac{W_0}{T} + R_t b_t + \frac{\phi_t}{\alpha_t}(R_t - \alpha_t)}{R(T)} \;\middle|\; \mathcal{F}^{t-1}\right]\right], \tag{13}$$

where the equality follows simply by rearrangement. Since $t \in \mathcal{H}^0$, invoking Lemma 1 guarantees that

$$\mathbb{E}\left[\frac{R_t}{R(T)} \;\middle|\; \mathcal{F}^{t-1}\right] = \mathbb{E}\left[\frac{\mathbf{1}\{P_t \le \alpha_t\}}{R(T)} \;\middle|\; \mathcal{F}^{t-1}\right] \le \mathbb{E}\left[\frac{\alpha_t}{R(T)} \;\middle|\; \mathcal{F}^{t-1}\right], \tag{14}$$

since the two mappings $(R_1,\ldots,R_T) \mapsto R(T)$ and $(R_1,\ldots,R_{t-1}) \mapsto \alpha_t \in \mathcal{F}^{t-1}$ are coordinate-wise nondecreasing, as required to apply Lemma 1. Since $\phi_t, \alpha_t$ are $\mathcal{F}^{t-1}$-measurable, equation (14) implies that the last term in the numerator of equation (13) is negative, and hence

$$L_t \le \mathbb{E}\left[\mathbb{E}\left[\frac{\frac{W_0}{T} + R_t b_t}{R(T)} \;\middle|\; \mathcal{F}^{t-1}\right]\right].$$

**Case 2.**  Now suppose that $t \notin \mathcal{H}^0$. Substituting $\psi_t \le \phi_t + b_t$ into the expression for $L_t$ yields

$$L_t \le \mathbb{E}\left[\mathbb{E}\left[\frac{\frac{W_0}{T} + R_t(\phi_t + b_t) - \phi_t}{R(T)} \;\middle|\; \mathcal{F}^{t-1}\right]\right]$$

$$= \mathbb{E}\left[\mathbb{E}\left[\frac{\frac{W_0}{T} + R_t b_t + \phi_t(R_t - 1)}{R(T)} \;\middle|\; \mathcal{F}^{t-1}\right]\right],$$

where the equality follows simply by rearrangement. Since $R_t \le 1$, we may infer that

$$L_t \le \mathbb{E}\left[\mathbb{E}\left[\frac{\frac{W_0}{T} + R_t b_t}{R(T)} \;\middle|\; \mathcal{F}^{t-1}\right]\right],$$

which is the same expression as the bound derived in Case 1.

**Combining both cases.** We complete the proof by combining the two cases. Using the same bound for $L_t$ in both cases yields

$$\mathbb{E}\left[\frac{V(T)+W(T)}{R(T)}\right] \leq \mathbb{E}\left[\frac{W_0 + \sum_t R_t b_t}{R(T)}\right]. \tag{15}$$

We now note that $b_t$ always equals $\alpha$, except for the very first rejection at time $\tau_1$, in which case it equals $\alpha - W_0$. Hence, we may have $\sum_t R_t b_t = \sum_t R_t \alpha - W_0 \mathbf{1}\{T \geq \tau_1\}$. Substituting this expression into the bound (15) yields

$$\mathbb{E}\left[\frac{V(T)+W(T)}{R(T)}\right] \leq \mathbb{E}\left[\frac{W_0 + \alpha R(T) - W_0 \mathbf{1}\{T \geq \tau_1\}}{R(T)}\right] \leq \alpha,$$

which completes the proof of the theorem.

# G    Proof of Theorem 3

Substituting the definitions of $V_u(T) = \sum_{t=1}^{T} u_t R_t \mathbf{1}\{t \in \mathcal{H}^0\}$ and the alpha-wealth

$$W(T) = W_0 + \sum_{t=1}^{T}(-\phi_t + R_t \psi_t),$$

we may use the tower property of conditional expectation to write

$$\mathbb{E}\left[\frac{V_u(T)+W(T)}{R_u(T)}\right] = \sum_t \underbrace{\mathbb{E}\left[\mathbb{E}\left[\frac{\frac{W_0}{T} + R_t(\psi_t + u_t\mathbf{1}\{t \in \mathcal{H}^0\}) - \phi_t}{R_u(T)}\,\bigg|\,\mathcal{F}^{t-1}\right]\right]}_{L_t}.$$

We tackle the above expression term by term, depending on whether or not $t \in \mathcal{H}^0$.

**Case 1.**   First suppose that $t \in \mathcal{H}^0$. Substituting $\psi_t \leq \frac{\phi_t}{u_t w_t \alpha_t} + u_t b_t - u_t$ into the expression for $L_t$ yields

$$L_t \leq \mathbb{E}\left[\mathbb{E}\left[\frac{\frac{W_0}{T} + R_t(\frac{\phi_t}{u_t w_t \alpha_t} + u_t b_t) - \phi_t}{R_u(T)}\,\bigg|\,\mathcal{F}^{t-1}\right]\right]$$

$$= \mathbb{E}\left[\mathbb{E}\left[\frac{\frac{W_0}{T} + R_t u_t b_t + \frac{\phi_t}{u_t w_t \alpha_t}(R_t - \alpha_t w_t u_t)}{R_u(T)}\,\bigg|\,\mathcal{F}^{t-1}\right]\right], \tag{16}$$

where the equality follows simply by rearrangement. Since $t \in \mathcal{H}^0$, by invoking Lemma 1, we may infer that

$$\mathbb{E}\left[\frac{R_t}{R_u(T)}\,\bigg|\,\mathcal{F}^{t-1}\right] = \mathbb{E}\left[\frac{\mathbf{1}\{P_t \leq \alpha_t w_t u_t\}}{R_u(T)}\,\bigg|\,\mathcal{F}^{t-1}\right] \leq \mathbb{E}\left[\frac{\alpha_t w_t u_t}{R_u(T)}\,\bigg|\,\mathcal{F}^{t-1}\right], \tag{17}$$

since the four mappings $(R_1, \dots, R_T) \mapsto R_u(T)$ and $(R_1, \dots, R_{t-1}) \mapsto \alpha_t, w_t, u_t$ are all coordinatewise nondecreasing, as required to apply Lemma 1. Since $\phi_t, \alpha_t, w_t, u_t$ are $\mathcal{F}^{t-1}$-measurable, equation (17) implies that the last term in the numerator of equation (16) is negative, and hence

$$L_t \leq \mathbb{E}\left[\mathbb{E}\left[\frac{\frac{W_0}{T} + R_t u_t b_t}{R_u(T)}\,\bigg|\,\mathcal{F}^{t-1}\right]\right].$$

**Case 2.**   Now suppose that $t \notin \mathcal{H}^0$. Substituting $\psi_t \leq \phi_t + u_t b_t$ into the expression for $L_t$ yields

$$L_t \leq \mathbb{E}\left[\mathbb{E}\left[\frac{\frac{W_0}{T} + R_t(\phi_t + u_t b_t) - \phi_t}{R_u(T)}\,\bigg|\,\mathcal{F}^{t-1}\right]\right]$$

$$= \mathbb{E}\left[\mathbb{E}\left[\frac{\frac{W_0}{T} + R_t u_t b_t + \phi_t(R_t - 1)}{R_u(T)}\,\bigg|\,\mathcal{F}^{t-1}\right]\right],$$

where the equality follows simply by rearrangement. Since $R_t \leq 1$, we may infer that

$$L_t \leq \mathbb{E}\left[\mathbb{E}\left[\frac{\frac{W_0}{T} + R_t u_t b_t}{R_u(T)}\,\bigg|\,\mathcal{F}^{t-1}\right]\right],$$

which is the same expression as the bound derived in Case 1.

**Combining both cases.** Finally, we combine the two cases. Using the same bound for $L_t$ in both cases, and exchanging the summation and expectation, we may conclude by definition of $b_t$ that

$$\mathbb{E}\left[\frac{V_u(T) + W(T)}{R_u(T)}\right] \leq \mathbb{E}\left[\frac{W_0 + \sum_t R_t u_t b_t}{R_u(T)}\right]. \tag{18}$$

We now note that $b_t$ always equals $\alpha$, except for the very first rejection at time $\tau_1$, in which case it equals $\alpha - \frac{W_0}{u_{\tau_1}}$. Hence, we may write $\sum_t R_t u_t b_t = \sum_t R_t u_t \alpha - W_0 \mathbf{1}\{T \geq \tau_1\}$. Substituting the above expression into the bound (18) yields

$$\mathbb{E}\left[\frac{V_u(T) + W(T)}{R_u(T)}\right] \leq \mathbb{E}\left[\frac{W_0 + \alpha R_u(T) - W_0 \mathbf{1}\{T \geq \tau_1\}}{R_u(T)}\right] \leq \alpha,$$

which completes the proof of the theorem.

# H   Proof of Theorem 4

Substituting the definitions of $V_u^\delta(T) = \sum_{t=1}^T \delta^{T-t} u_t R_t \mathbf{1}\{t \in \mathcal{H}^0\}$ and the alpha-wealth

$$W(T) = W_0 \delta^{T-\min\{\tau_1, T\}} + \sum_{t=1}^T \delta^{T-t}(-\phi_t + R_t \psi_t),$$

we may use the tower property to write

$$\mathbb{E}\left[\frac{V_u^\delta(T) + W(T)}{R_u^\delta(T)}\right] = \sum_t \mathbb{E}\left[\underbrace{\mathbb{E}\left[\frac{\frac{W_0}{T}\delta^{T-\min\{\tau_1,T\}} + \delta^{T-t}R_t(\psi_t + u_t\mathbf{1}\{t \in \mathcal{H}^0\}) - \delta^{T-t}\phi_t}{R_u^\delta(T)}\;\middle|\;\mathcal{F}^{t-1}\right]}_{L_t}\right].$$

We tackle the above expression term by term, depending on whether or not $t \in \mathcal{H}^0$.

**Case 1**   First suppose that $t \in \mathcal{H}^0$. Substituting $\psi_t \leq \frac{\phi_t}{u_t w_t \alpha_t} + u_t b_t - u_t$ into the expression for $L_t$ yields

$$L_t \leq \mathbb{E}\left[\mathbb{E}\left[\frac{\frac{W_0}{T}\delta^{T-\min\{\tau_1,T\}} + \delta^{T-t}R_t(\frac{\phi_t}{u_t w_t \alpha_t} + u_t b_t) - \delta^{T-t}\phi_t}{R_u^\delta(T)}\;\middle|\;\mathcal{F}^{t-1}\right]\right]$$

$$= \mathbb{E}\left[\mathbb{E}\left[\frac{\frac{W_0}{T}\delta^{T-\min\{\tau_1,T\}} + \delta^{T-t}R_t u_t b_t + \delta^{T-t}\frac{\phi_t}{u_t w_t \alpha_t}(R_t - \alpha_t w_t u_t)}{R_u^\delta(T)}\;\middle|\;\mathcal{F}^{t-1}\right]\right], \tag{19}$$

where the equality follows simply by rearrangement. Since $t \in \mathcal{H}^0$, by invoking Lemma 1, we may infer that

$$\mathbb{E}\left[\frac{R_t}{R_u^\delta(T)}\;\middle|\;\mathcal{F}^{t-1}\right] = \mathbb{E}\left[\frac{\mathbf{1}\{P_t \leq \alpha_t w_t u_t\}}{R_u^\delta(T)}\;\middle|\;\mathcal{F}^{t-1}\right] \leq \mathbb{E}\left[\frac{\alpha_t w_t u_t}{R_u^\delta(T)}\;\middle|\;\mathcal{F}^{t-1}\right], \tag{20}$$

since the four mappings $(R_1, \ldots, R_T) \mapsto R_u^\delta(T)$ and $(R_1, \ldots, R_{t-1}) \mapsto \alpha_t, w_t, u_t$ are coordinatewise nondecreasing, as required to apply Lemma 1. Since $\phi_t, \alpha_t, w_t, u_t$ are $\mathcal{F}^{t-1}$-measurable, equation (20) implies that the last term in the numerator of equation (19) is negative, and hence

$$L_t \leq \mathbb{E}\left[\mathbb{E}\left[\frac{\frac{W_0}{T}\delta^{T-\min\{\tau_1,T\}} + \delta^{T-t}R_t u_t b_t}{R_u^\delta(T)}\;\middle|\;\mathcal{F}^{t-1}\right]\right].$$

**Case 2**   Next, suppose that $t \notin \mathcal{H}^0$. Substituting $\psi_t \leq \phi_t + u_t b_t$ into the expression for $L_t$ yields

$$L_t \leq \mathbb{E}\left[\mathbb{E}\left[\frac{\frac{W_0}{T}\delta^{T-\min\{\tau_1,T\}} + \delta^{T-t}R_t(\phi_t + u_t b_t) - \delta^{T-t}\phi_t}{R_u^\delta(T)}\;\middle|\;\mathcal{F}^{t-1}\right]\right]$$

$$= \mathbb{E}\left[\mathbb{E}\left[\frac{\frac{W_0}{T}\delta^{T-\min\{\tau_1,T\}} + \delta^{T-t}R_t u_t b_t + \delta^{T-t}\phi_t(R_t - 1)}{R_u^\delta(T)}\;\middle|\;\mathcal{F}^{t-1}\right]\right],$$

where the equality follows simply by rearrangement. Since $R_t \leq 1$, we may infer that

$$L_t \leq \mathbb{E}\left[\mathbb{E}\left[\frac{\frac{W_0}{T}\delta^{T-\min\{\tau_1,T\}} + \delta^{T-t}R_t u_t b_t}{R_u^\delta(T)}\;\middle|\;\mathcal{F}^{t-1}\right]\right],$$

which is the same expression as the bound derived in Case 1.

**Combining Cases 1 and 2.** Using the same bound for $L_t$ in both cases, and exchanging the summation and expectation, we may conclude by definition of $b_t$ that

$$\mathbb{E}\left[\frac{V_u^\delta(T) + W(T)}{R_u^\delta(T)}\right] \le \mathbb{E}\left[\frac{W_0\delta^{T-\min\{\tau_1,T\}} + \sum_t \delta^{T-t}R_t u_t b_t}{R_u^\delta(T)}\right]. \tag{21}$$

We now note that $b_t$ always equals $\alpha$, except for the very first rejection at time $\tau_1$, in which case it equals $\alpha - \frac{W_0}{u_{\tau_1}}$. Hence, we may write

$$\sum_t \delta^{T-t}R_t u_t b_t = \sum_t \delta^{T-t}R_t u_t \alpha - \delta^{T-\tau_1}W_0 \mathbf{1}\{T \ge \tau_1\}$$

$$= \alpha R_u^\delta(T) - \delta^{T-\tau_1}W_0 \mathbf{1}\{T \ge \tau_1\}.$$

Resubstituting this expression into bound (21) yields

$$\mathbb{E}\left[\frac{V_u^\delta(T) + W(T)}{R_u^\delta(T)}\right] \le \mathbb{E}\left[\frac{W_0\delta^{T-\min\{\tau_1,T\}} + \alpha R_u^\delta(T) - W_0\delta^{T-\tau_1}\mathbf{1}\{T \ge \tau_1\}}{R_u^\delta(T)}\right] \le \alpha,$$

where the last inequality follows by verifying that it holds in the three cases

$$\{T < \tau_1 = \infty, R_u^\delta(T) = 0\}, \quad \{T \ge \tau_1, R_u^\delta(T) < 1\}, \quad \text{and} \quad \{T \ge \tau_1, R_u^\delta(T) \ge 1\}$$

separately. This completes the proof of the theorem.

## Footnotes

[1]The code for reproducing all experiments in this paper and producing graphs is publicly available at https://github.com/fanny-yang/OnlineFDRCode.