[Reviews · NeurIPS 2017]

Reviewer 1



A very well written contribution on an important topic. I find that each of the 3 contributions in the paper could justify a paper for themselves: The GAI++ algorithm is a simple generalization of the GAI algorithms, that settle an interesting open question regarding FDR versus mFDR control. The weighted-online FDR control is a nice extension of online-FDR. Most importantly, in my opinion, is the introduction of the mem-FDR which is a natural adaptation of the exponentially weighted moving average to FDR control. I am surprised this has not been suggested previously, and is a natural and important error criterion. It is for this last contribution that I expect this paper to have a high impact.

Reviewer 2



This paper unifies and extends concepts related to the online false discovery rate (FDR) control. This is a recent trendy setting where null hypotheses are coming in a sequential manner over time and the user should make an (irrevocable) decision about rejecting (or not) the null coming at each time. The aim is to make a sequential inference so that the expected proportion of errors among the rejected nulls is bounded by a prescribed value $\alpha$ (at any stopping time). My opinion is that this paper is of very high quality: the writing is good and easy to follow, the motivation is clear and interesting, the new procedures and concepts are clever and the proofs are really nicely written. My only concern is that the strongest innovation (decaying memory procedure) is maybe not sufficiently well put forward in the main paper. It only comes at the last 1.5 pages and there is no illustration for Alpha-death recovery. This concept seems however more important than the double-weighting of Section 4 that can be put in appendix safely (more standard). I would also suggest to put Section D.3 of the supplement in the main paper (illustration of Piggybacking). Here are some other comments and typos : line 58 "will equal 1" (only with high probability); line 131 "mst"; line 143 "keeping with past work"; Notation "W(t)" I suggest to use W_t (if you agree); Lemma 1 f_i is not F_i measurable but just measurable (right?); Section D.1, line 469-470 :" mu_j follows 0" should read "mu_j=0"; line 528 : equation (13) should read equation (10); line 529-530 : -phi_t (R_t-1) should read +phi_t (R_t-1).